# Advancements in Research and Treatment Applications of Patient-Derived Tumor Organoids in Colorectal Cancer

**DOI:** 10.3390/cancers16152671

**Published:** 2024-07-26

**Authors:** Denise van der Graaff, Sofie Seghers, Pieterjan Vanclooster, Christophe Deben, Timon Vandamme, Hans Prenen

**Affiliations:** 1Department of Medical Oncology, University Hospital Antwerp, 2650 Edegem, Belgium; 2Center for Oncological Research (CORE), University of Antwerp, 2610 Wilrijk, Belgium

**Keywords:** patient-derived tumor organoids, colorectal cancer, cancer, precision medicine, organoids, drug resistance

## Abstract

**Simple Summary:**

Colorectal cancer (CRC) is a major global health issue, causing many deaths each year. The current treatments often face challenges due to the cancer’s resistance to drugs, making it hard to treat effectively. This research focuses on using patient-derived tumor organoids (PDTOs), which are small, three-dimensional models grown from a patient’s tumor, to better understand and predict how CRC responds to treatments. PDTOs mimic the actual tumor environment closely, making them more reliable than traditional models. By studying PDTOs, researchers hope to find new ways to use existing drugs, understand why resistance happens, and develop better treatment plans tailored to individual patients. This approach could lead to significant advancements in how CRC is managed, potentially improving survival rates and patient outcomes.

**Abstract:**

Colorectal cancer (CRC) remains a significant health burden globally, being the second leading cause of cancer-related mortality. Despite significant therapeutic advancements, resistance to systemic antineoplastic agents remains an important obstacle, highlighting the need for innovative screening tools to tailor patient-specific treatment. This review explores the application of patient-derived tumor organoids (PDTOs), three-dimensional, self-organizing models derived from patient tumor samples, as screening tools for drug resistance in CRC. PDTOs offer unique advantages over traditional models by recapitulating the tumor architecture, cellular heterogeneity, and genomic landscape and are a valuable ex vivo predictive drug screening tool. This review provides an overview of the current literature surrounding the use of PDTOs as an instrument for predicting therapy responses in CRC. We also explore more complex models, such as co-cultures with important stromal cells, such as cancer-associated fibroblasts, and organ-on-a-chip models. Furthermore, we discuss the use of PDTOs for drug repurposing, offering a new approach to identify the existing drugs effective against drug-resistant CRC. Additionally, we explore how PDTOs serve as models to gain insights into drug resistance mechanisms, using newer techniques, such as single-cell RNA sequencing and CRISPR-Cas9 genome editing. Through this review, we aim to highlight the potential of PDTOs in advancing our understanding of predicting therapy responses, drug resistance, and biomarker identification in CRC management.

## 1. Introduction

Colorectal cancer (CRC) is currently the third most prevalent cancer globally, with an estimated 1.9 million new cases in 2020, and represents the second leading cause of cancer-related deaths, accounting for over 900,000 deaths annually [1]. Metastases are common, as up to 30% of new diagnoses present with metastatic disease and up to 50% of patients initially diagnosed with localized disease eventually develop metastases [2].

Even with the recent developments in therapeutic approaches, including the introduction of potent new drugs targeting receptor tyrosine kinases such as epidermal growth factor receptor (EGFR) inhibitors and immune checkpoint inhibitors, a substantial proportion of CRC patients experience relapse during the course of treatment [3]. Resistance to systemic antineoplastic agents, either intrinsic or acquired, remains a critical challenge in the management of CRC and significantly contributes to patient morbidity and mortality. Consequently, drug resistance presents a significant challenge, compromising the efficacy of treatment strategies for individuals with colorectal cancer. Additionally, there is a growing recognition of the importance of personalized medicine approaches in CRC management, wherein treatment decisions are tailored to the individual patient based on their unique tumor profile. Therefore, the development of innovative screening tools to tailor patient-specific treatment is pivotal to improving patient care.

Patient-derived tumor organoids (PDTOs) are miniature three-dimensional (3D), self-organizing tissue structures derived from patient tumor samples, which retain several characteristics from the tumor of origin in vitro [4]. PDTOs offer unique advantages over traditional models by recapitulating the tumor architecture, cellular heterogeneity, genomic landscape, and tumor microenvironment interactions. This enhanced fidelity allows for PDTOs to provide a more authentic representation of tumor biology compared to conventional models. The use of PDTOs can significantly enhance research aimed at optimizing our understanding of CRC. Moreover, PDTOs are a promising advancement in clinical cancer treatment as they can be used as an ex vivo predictive drug screening tool. As such, organoids harbor the potential to revolutionize cancer research and clinical care, offering new opportunities for precision medicine and therapeutic development.

The aim of this review is to explore the application of PDTOs as an instrument for the understanding of CRC and predicting therapy responses in CRC, hereby reducing CRC morbidity and mortality.

## 2. Cultivating Complexity: From Traditional Models to Advanced Organoids

The evolution of cellular models has progressed from basic 2D cultures to advanced 3D organoids, overcoming the limitations of traditional methods and ethical concerns of in vivo models. PDTOs offer accurate representations of human physiology and heterogeneity, and co-culturing them with various cell types enables the study of complex cell interactions.

### 2.1. Preclinical Landscape: Navigating Animal Models, 2D Cultures, and Patient-Derived Xenografts

To date, several models of CRC are available for preclinical research in CRC. We will briefly discuss the most commonly used in vitro and in vivo CRC models.

The first animal models developed were the carcinogen-induced rodent models (CIMs), such as mice treated with azoxymethane and dextran sulfate sodium, which induce colitis-associated cancer [5,6]. While useful for studying carcinogens and colitis-related malignancies, CIMs mainly represent early-stage CRC and rarely show invasive or metastatic disease [7]. Genetically engineered mouse models (GEMMs) are useful for studying specific molecular pathways in the development of CRC and evaluating new treatments [8]. However, GEMMs lack validated models for metastasized CRC [9,10]. Moreover, GEMMs do not reflect the genetic diversity seen in humans [9]. Finally, a patient-derived tumor xenograft (PDTX) involves implanting human tumor samples into immunodeficient mice, thus representing tumor heterogeneity and aiding biomarker discovery and treatment evaluation [11,12]. To the disadvantage of all animal models, species-specific differences and ethical concerns limit the translation of animal models to human medicine [13,14]. 

In vitro, cancer cell lines (2D cultures) from human CRC tissues provide a cost-effective method for studying treatment responses in a controlled environment. However, they lack the complexity of 3D tumors, interactions with surrounding tissues, and immune cells, reducing their relevance to human CRC [15,16].

All of the examples above have their own limitations, most importantly the lack of heterogeneity of tumor tissue in animal models and the absence of 3D complexity, limiting their translational relevance.

### 2.2. Generating and Culturing CRC PDTOs

Given the limitations of existing animal and in vitro 2D models in accurately replicating human CRC, PDTOs have been developed. We will briefly discuss the technical aspects involved in the development of PDTOs. 

Generating ex vivo PDTOs entails various protocols, which have notable similarities. An overview of a standard protocol is depicted in Figure 1. Fundamental steps include the collection of patient-derived primary cancer cells, digestion into single cells, centrifugation to form a cell pellet, and the subsequent encapsulation of cells either in domes by incorporating basal membrane extract (BME) or in suspension culture. 

Primary cancer cells can originate from diverse sources, including resection specimens, biopsies, or malignant ascites. Resection specimens or biopsies, being the most frequently used techniques, can be obtained from either primary tumors or metastatic sites. These specimens may be obtained from patients that are either pretreated or treatment-naïve. Notably, the derivation of PDTOs from malignant ascites offers certain advantages over biopsies or resections, such as cost-efficiency and reduced invasiveness; however, it is limited to cases of metastatic CRC [17]. An innovative approach involves generating PDTOs from circulating tumor cells, a technique warranting further exploration [18].

In solid tissue specimen processing, various protocols are available, with the method outlined by Sato et al. [19] being widely adopted. While these protocols may differ in nuances, they generally follow a similar sequence of steps. Initially, the tissue is finely minced and subjected to enzymatic treatment to aid in digestion, leading to the isolation of (near) single cells. Following enzymatic treatment, the isolated cells are either encapsulated within domes composed of BME, such as Matrigel or Cultrex, or maintained in suspension cultures. Subsequently, a tumor-specific medium is introduced, often consisting of advanced Dulbecco’s Modified Eagle Medium/Ham’s F-12 (DMEM/F12) supplemented with growth factors tailored to support tumor cell growth and viability. The composition of the medium varies across protocols, with common additives including R-Spondin, noggin, B27, N-acetyl-cysteine (NAC), nicotinamide, A83-01, and SB202190 (an overview can be found in Appendix A). In contrast to various other tumor types, Wnt signaling does not seem to play a critical role in CRC medium. Interestingly, when Wnt is removed from the growth medium, healthy colon cells are selectively eliminated while the cancer cells are preserved [20]. Notably, Tan et al. successfully developed a growth factor-reduced medium devoid of noggin, R-Spondin, or Wnt supplementation [21]. On the other hand, fetal bovine serum (FBS) is frequently added to CRC medium; however, due to its substantial inter-batch variability, which can lead to discrepancies between experiments, its inclusion is not recommended for PDTO culturing [22]. Medium replacement is typically performed every 2–3 days. Every 1–2 weeks, PDTOs can be passaged, often using trypLE to dissociate the PDTOs into single cells or small cell clusters, thereby facilitating exponential growth of the PDTO culture.

The success rate of PDTO generation from CRC specimens ranges between 50 and 100% (see overview in Appendix A) [23]. Notably, Li et al. identified certain tumor biology markers associated with lower success rates for PDTO development, including microsatellite instability (MSI), BRAF mutations, a mucinous subtype, and poorly differentiated tumors [24]. Moreover, biopsies often have lower cell counts than resection specimens, leading to a lower success rate [25,26]. Tumor biology and sampling type should thus be considered critical determinants when establishing PDTO cultures.

### 2.3. Beyond the Basics: Exploring More Complex Models

CRC is profoundly shaped by its TME, comprising a multitude of components, including, amongst others, fibroblasts, immune cells, and endothelial cells. PDTOs alone, as an in vitro model, inadequately capture the complexity of these interactions with the TME. Furthermore, critical therapeutic modalities such as immunotherapies and angiogenesis inhibitors cannot be accurately assessed in a model of isolated cancer cells. Consequently, efforts have been made to incorporate various cell types alongside CRC PDTOs, including cancer-associated fibroblasts (CAFs), endothelial cells, and immune cells. This combination of PDTOs with one or more additional cell types is often referred to as “assembloids” [27].

Different studies include the interplay between CRC PDTOs and CAFs. In one study, a co-culture system was developed using CRC PDTOs and immortalized CAFs, which spontaneously organized into structures resembling aggressive mesenchymal-like colon cancer. Using single-cell RNA sequencing (scRNAseq), it was observed that CAFs elicited a partial epithelial-to-mesenchymal transition in a subset of cancer cells, similar to the characteristics observed in the mesenchymal-like consensus molecular subtype 4. These co-cultures exhibited histological, biophysical, and immunosuppressive features typical of this cancer subtype, providing a platform to investigate the mechanisms of immunosuppression and to test therapeutic interventions targeting the interplay between CAFs and cancer cells [28]. Similarly, Farin et al. underscore the indispensability of CAFs for a faithful representation of molecular subtypes and therapy responses in CRC. Their study introduced a matched organoid–stroma biobank, housing paired PDTOs and CAFs from thirty patients. They observed that under standard conditions, CRC PDTOs tend to lose their original subtype, but co-culturing them with CAFs could partially restore this subtype. Moreover, significant differences in therapy responses were noted when CRC PDTOs were co-cultured with CAFs. Notably, CAFs were found not only to confer resistance in a general sense but also in a patient- and therapy-dependent manner [29]. In another study, focusing on identifying a predictive biomarker for regorafenib resistance, triple co-cultures were employed comprising CRC PDTOs (50%), CAFs (10%), and endothelial cells (40%) [30]. The inclusion of CAFs was motivated by their known influence on resistance mechanisms, while endothelial cells were incorporated due to the anti-angiogenic properties of regorafenib. Leveraging scRNAseq, the study successfully modeled the regorafenib response observed in patients. They demonstrated that the suggested biomarker MIR652-3p controls resistance by impairing autophagy and inducing a transition from neo-angiogenesis to vessel co-option [30]. In another recent study, CRC PDTOs were co-cultured with human monocyte-derived macrophages. When macrophages were introduced, they acquired an immunosuppressive and pro-tumorigenic gene expression profile similar to in vivo observations, including the hallmark induction of SPP1. While CAFs alone could not induce the SPP1+ state in macrophages, their presence with PDTOs enhanced SPP1+ macrophage populations [31]. 

Beyond stromal cells, a diverse array of immune cells can be integrated into CRC PDTOs for various investigative purposes, including testing immunotherapy or evaluating adoptive T-cell therapy. In a recent study, CRC PDTOs were co-cultured with tumor-infiltrating lymphocytes (TILs), revealing the potential of autologous PDTOs to screen for TIL reactivity, thereby facilitating personalized adoptive cell therapy [32]. Another study demonstrated that triple co-cultures with T-cells, CAFs, and CRC PDTOs are a good model to evaluate the anti-tumor effect of two bi-specific T-cell engagers (CEA-TBC and CEACECAM5-TBC) [33]. Sui et al. developed tumor organoids sourced from MSI-high CRC patients undergoing PD-1 blockade therapy and co-cultivated them with either TILs or T cells derived from peripheral blood mononuclear cells (PBMCs). Their findings revealed that patients experiencing local inflammatory states during treatment displayed a heightened likelihood of disease progression and poorer progression-free survival rates [34]. Another study demonstrates the feasibility of using MSI high CRC PDTO co-cultures to expand tumor-reactive T cells from PBMCs, offering a promising approach for personalized immunotherapy. This platform not only provides insights into tumor-immune interactions but also paves the way for developing patient-specific T cell-based therapies and combinatory treatment strategies [35]. 

Organoids-on-a-chip technology has emerged as a powerful tool for cancer research, combining the advantages of patient-derived tumor organoids (PDTOs) and organ-on-a-chip (OoC) systems. This hybrid model replicates key features of the TME, including multicellular structures, tissue–tissue interfaces, chemical gradients, vascular perfusion, and mechanical properties. By integrating PDTOs with OoC technology, researchers have created physiologically relevant models that enhance our understanding of tumor biology, improve drug screening processes, and advance personalized medicine [36,37]. Despite these advances, further improvements are needed to address limitations, such as the complexity of 3D structures, technical robustness, and the integration of a complete tumor microenvironment (TME). With continued development, we anticipate that organoids on a chip will significantly advance both fundamental and translational cancer research, as well as enhance personalized medicine in the near future.

To conclude, co-cultures with various cells and organoids-on-a-chip technology can improve the PDTO culture to better reflect the cellular heterogeneity and improve testing of various compounds, such as immunotherapy and angiogenesis inhibitors.

## 3. CRC PDTOs as Innovative Instruments in the Preclinical Setting

CRC PDTOs prove invaluable in preclinical research, facilitating the exploration of resistance mechanisms and the development or repurposing of therapeutic strategies to combat or prevent resistance. Furthermore, their application extends to identifying predictive biomarkers, thereby optimizing personalized medicine strategies.

### 3.1. Insights into Resistance: Illuminating Mechanisms with Innovative Techniques

PDTOs can serve as an important model for gaining insights into drug resistance mechanisms, using techniques such as proteotranscriptomics, scRNAseq, and CRISPR-Cas9 genome editing. A recent study utilizing proteotranscriptomic analysis of CRC PDTOs highlights their potential in elucidating resistance mechanisms. This study provides insights into sensitivity to oxaliplatin and palbociclib, focusing on the interplay between genomic alterations, transcriptomic profiles, and proteomic signatures in drug responses. The integration of proteomic and transcriptomic data helps reveal the molecular basis of drug resistance [38]. Recent advances in scRNAseq applied to CRC PDTOs have significantly enhanced the characterization of these models and deepened our understanding of the TME, its impact on the drug response, and biomarker discovery. For instance, Wang et al. demonstrated that organoids derived from CRC patients accurately reflect the gene expression profiles of in vivo tumor cells. They also found that while organoids from normal tissues exhibited some tumor-like transcriptomic features, they preserved normal genomic characteristics, including CNVs, point mutations, and DNA methylation patterns [39]. Additionally, Strating et al. and Li et al. employed scRNAseq with PDTO co-cultures involving CAFs and macrophages, respectively, to investigate the TME effects on cancer cells [28,31]. Moreover, Hedayat et al. used scRNAseq to identify a biomarker linked to regorafenib resistance [30]. Collectively, these studies highlight the impressive impact of scRNAseq in providing insights into tumor–stroma interactions, drug resistance mechanisms, and the fidelity of organoid models in cancer research.

In CRC research, the emergence of CRISPR-Cas9 technology has marked a significant turning point in our ability to unravel resistance mechanisms, particularly within the context of PDTOs [40]. CRISPR-Cas9 enables precise genome editing, facilitating the interrogation of key genes and pathways implicated in resistance development [41]. While early investigations primarily relied on targeted CRISPR screens with a limited number of genes, the landscape dramatically shifted with the arrival of groundbreaking genome-scale studies, exemplified by the work of Ringel et al. and other research groups [42,43]. These comprehensive projects have revolutionized our approach by enabling the genome-wide interrogation of resistance mechanisms within CRC, all while streamlining experimental procedures through innovations such as the reduction in cell numbers and integrating unique molecular identifiers [44], thus enhancing screening accuracy. Despite the remarkable progress facilitated by these technological advancements, challenges remain. Predicting the functionality of guide RNAs (gRNAs) and managing clonal drift during prolonged screening are ongoing obstacles [44]. 

Notable discoveries include the use of CRISPR dropout screens as a powerful tool for uncovering critical pathways implicated in CRC resistance, with cholesterol synthesis emerging as a promising target for therapeutic intervention [42]. Furthermore, the targeted manipulation of key genes in CRC PDTOs, such as SMAD4 [45], NF-1 [46], MIEF2 [47], and RNF43 [48], has revealed their roles in mediating sensitivity or resistance to various therapeutic modalities, offering valuable insights for developing precision medicine strategies.

Beyond conventional gene knockouts, the continuous refinement of CRISPR techniques, including the advent of base editing, has created opportunities for investigating tumorigenesis and resistance mechanisms in CRC. The work by Clevers’ group, for instance, showcases the transformative potential of CRISPR-Cas9 technology in recapitulating CRC tumorigenesis within organoid models through the precise modulation of multiple oncogenic drivers [49].

### 3.2. Investigating Intratumoral Heterogeneity and Resistant Subclones

Resistance to treatment is often driven by intratumoral heterogeneity and the presence of resistant subclones within tumors and metastatic sites [50]. These subclones may preexist or emerge following therapeutic interventions [51]. Our understanding of the extent of this intratumoral heterogeneity and the presence of resistant subclones in CRC remains insufficient. However, recent advancements have provided new insights into this issue. 

A pioneering effort has been the establishment of a living biobank comprising 50 matched PDTOs from 25 individuals with CRC and liver metastases. Through comprehensive multi-omics characterization, including histopathology, genomics, transcriptomics, and scRNAseq, coupled with drug screenings, this biobank has effectively captured interpatient heterogeneity and revealed diverse responses to both monotherapy and combination therapies [52]. Moreover, Song et al. demonstrated significant intratumoral heterogeneity by establishing 15 single-clone PDTO lines from four tumor biopsies of a single CRC patient, each exhibiting variable genomic and phenotypic alterations along with distinct drug responses [53]. Similarly, the laboratory of Hans Clevers reported substantial differences in methylation, transcriptome states, and therapeutic responses among single-clone organoids derived from the same patient [54]. 

Notably, resistant subclones are implicated not only in resistance to systemic therapies but also in radioresistance. A recent study elucidated the inherent radioresistance conferred by preexisting resistant subclones, as demonstrated by a comparative analysis of single-cell whole-genome karyotyping between irradiated and unirradiated rectal PDTOs. This study underscored that radioresistance arises from resilient subpopulations that persist or expand rather than being induced de novo by irradiation [55]. 

In summary, the current paradigm of precision oncology in metastatic CRC is limited by its reliance on single lesion analysis. The inclusion of multiple-lesion PDTOs is needed to comprehensively mimic the landscape of intertumoral heterogeneity, which profoundly influences drug sensitivity screening. Moreover, identifying resistant subclones can be beneficial in response prediction. 

### 3.3. Drug Discovery Redefined: Drug Repurposing and Synergy Screening

In the quest for improved treatment strategies, drug repurposing and synergy screening have emerged as crucial tools. Leveraging existing medications, with their known safety profiles and pharmacological properties, accelerates therapeutic development. PDTOs play a pivotal role in this process, enabling efficient screening for new uses or combinations of drugs.

Numerous studies have investigated drug repurposing for CRC. Mao et al. conducted a screening of 335 drugs, identifying 34 compounds with anti-CRC properties. Subsequent in vivo validation confirmed the efficacy of fedratinib, trametinib, and bortezomib against CRC [56]. Another study focused on high-risk colorectal adenomas, which are precancerous lesions currently treated with endoscopic resection. Through the high-throughput screening of 139 compounds, researchers identified potential candidates such as metformin and panobinostat for growth inhibition in high-risk colorectal adenomas, offering promise in preventing progression to CRC [57]. Additionally, research utilizing CRC PDTOs revealed that itraconazole exhibited inhibitory effects on CRC, warranting further exploration through clinical trials [58].

Combination therapies enable the concurrent targeting of heterogeneous subclone populations within tumors, each harboring unique vulnerabilities. This approach enhances the efficacy of systemic therapies by extending their coverage across the tumor landscape [59]. Conducting synergy screening on CRC PDTOs can optimize combination strategies by identifying synergistic drug pairs that exhibit enhanced efficacy when used together. Mertens et al. screened a library of 414 drugs on various CRC PDTOs, aiming to transition the cytostatic effect induced by MEK and EGFR inhibitors to a more cytotoxic effect. Their analysis yielded 37 hits, highlighting microtubule inhibitors as promising candidates when combined with MEK/EGFR inhibitors. Notably, the combination of vinorelbine and pan-HER/MEK inhibitors demonstrated high efficacy, prompting the initiation of a clinical trial (RASTRIC—EudraCT: 2019-004987-23) to validate these findings [60]. Cholesterol biosynthesis inhibitors, such as statins and zoledronate, have also been shown to enhance the anticancer effect of 5-fluoro-uracil (5-FU) in CRC PDTOs and in vivo mouse models [42]. Lee et al. validated the postulated lethal synergistic effect of two anti-metabolic drugs, phenformin and 2-deoxy-D-glucose, in four CRC PDTO lines [61]. 

In summary, drug repurposing and synergy screening, facilitated by high-throughput drug screening on PDTOs, are pivotal in advancing CRC treatment. These approaches not only expedite the identification of effective therapies but also enhance the precision of combination strategies.

### 3.4. Pioneering New Therapies

Ninety percent of phase 1 clinical trials currently face failure [62], underscoring the significant translational challenges from preclinical models to clinical settings (Section 2.1). Addressing this, PDTOs represent a substantial breakthrough in drug discovery and translation to the clinical setting, faithfully recapitulating the heterogeneity of human cancer cells [63]. 

For instance, a pioneering study establishing a biobank of 22 CRC PDTOs conducted high-throughput drug screenings, identifying potential molecular markers by integrating drug response data with genomic profiles [64]. Integrative analyses like these offer valuable insights into the molecular basis of drug sensitivity and resistance.

As another example, recent investigations have shown promising outcomes with compounds like ML264, a KLF5 inhibitor, which enhanced sensitivity in CRC models when combined with standard chemotherapeutic agents, like oxaliplatin [65]. Additionally, inhibition of the hedgehog pathway using compounds like AY9944 and GANT61 demonstrated significant decreases in CRC PDTO viability when combined with conventional therapies like 5-FU or irinotecan, suggesting the potential of hedgehog inhibitors as novel therapeutic targets [66].

In summary, these advancements in PDTO-based drug discovery highlight the technology’s potential to accelerate the development of more effective and personalized therapies for CRC patients.

### 3.5. Discovery of Predictive Biomarkers for Personalized Medicine in CRC 

PDTOs offer a promising avenue for elucidating predictive biomarkers in CRC through analyses of their molecular and genetic characteristics. These biomarkers encompass a spectrum of factors such as gene mutations, expression patterns, and alterations in signaling pathways crucial for drug sensitivity or resistance. Numerous examples underscore the potential of PDTO-derived predictive biomarkers across various therapeutic modalities for CRC. 

Differential gene expression analyses using drug screening assays on CRC PDTOs have identified gene signatures with predictive capabilities in CRC. For instance, the Drug Resistant Score Model (DRSM), comprising five genes *CACNA1D*, *CIITA*, *PFN2*, *SEZ6L2*, and *WDR78,* has showed promise to forecast sensitivity to 5-FU in CRC in the initial validation [67]. Still, achieving higher predictive accuracy remains a clinical imperative. Similarly, 18 predictive signatures for oxaliplatin sensitivity have been discovered by using molecular profiling of the transcriptomes of CRC PDTOs [68]. Likewise, Pfohl et al. discovered the SFAB-signature, consisting of *SMAD4*, *FBXW7*, *ARID1A*, and *BMPR2*, which predicts sensitivity to MEK inhibitors independent of the RAS and BRAF status. This signature demonstrated robust positive predictive values for CRC, particularly for cobimetinib and selumetinib [45].

Beyond differential gene expression, other innovative approaches in combination with PDTOs have unveiled additional predictive biomarkers in CRC. Hedayat et al. [30] identified MIR652-3p as a biomarker indicative of clinical benefit from regorafenib, employing large-scale microRNA expression analyses and mechanistic validation through triple co-cultures, with CRC PDTOs, CAFs, and endothelial cells, and scRNAseq. Another study employing a CRISPR-mediated knockout panel of RASGAPs in CRC PDTOs highlighted the significance of NF1 loss in enhancing tumor growth under low EGF signaling conditions [46]. Post et al. underscore the potential utility of a pretreatment NF-1 assessment prior to MAPK pathway inhibitor therapy [46]. Moreover, network-based machine learning holds promise in predicting anticancer drug efficacy. Kong et al. [69] utilized this approach to identify predictive biomarkers for 5-FU in CRC, revealing correlations between the expression levels of components within the “activation of BH3-only proteins” pathway and CRC PDTO responses to 5-FU. However, validation of these biomarkers remains needed for clinical translation.

In summary, PDTOs are valuable for identifying predictive biomarkers in CRC through molecular and genetic analyses. These biomarkers, encompassing gene mutations, expression patterns, and signaling pathway alterations, can indicate drug sensitivity or resistance. However, clinical validation is essential to ensure their accuracy and utility in treatment.

## 4. From Bench to Bedside: Clinical Insights and Applications

The clinical treatment of CRC faces challenges due to the heterogeneity of tumor responses to systemic treatments, like chemotherapy, immunotherapy, or targeted therapies. Currently, a standardized sequence of treatment options is generally applied to every patient based on their disease stage. While targeted therapy efficacy can be partially predicted through genomic profiling [70], the response to chemotherapies remains unpredictable in clinical practice, leading to potential tumor progression and decreased quality of life [71,72]. Therefore, the main clinical promise of PDTOs lies in their potential for personalized medicine. As mentioned above, PTDOs can mimic on various levels the parental tumor tissue [64,73]. Subsequently exposing this tumor tissue to potential treatments in vitro, they can predict the patient’s specific tumor response [74].

### 4.1. Assessing Predictive Value in Clinical Therapy Response 

Assessing the predictive value of PDTOs in the clinical therapy response is essential before their clinical implementation. Currently, the available data predominantly consist of retrospective analyses and case studies. However, in a recent systematic review, CRC PDTO drug screening demonstrated a mean positive predictive value (PPV) of 68% and a negative predictive value (NPV) of 78% across various treatment regimens, including chemotherapy, targeted therapy, and radiotherapy [23].

Numerous case studies suggest the feasibility of generating PDTOs for drug screening, showing concordance between ex vivo drug screening results and patient responses [74,75,76]. However, these data are limited to single cases without controls, lacking statistical power to draw robust conclusions about accuracy or prognostic impact.

Likewise, retrospective studies, in which PDTO responses are compared to clinical patient responses mostly based on follow-up data, generally demonstrate promising results. The potential of the PDTOs’ response rates in vitro to predict in vivo treatment responses was already demonstrated in a post hoc study in 2018, in which PDTOs from metastasized tumors, including CRC, were treated with subsequent-line modalities ex vivo, paralleling clinical treatment. The study showed a correlation between ex vivo and in vivo treatment responses to chemotherapy and targeted therapies, with a PPV of 88% and an NPV of 100% [25]. Wang et al. reported similar findings in stage IV CRC. Clinical response to chemotherapy could be predicted by the PDTO response with an accuracy of 80% [77]. They subsequently developed a model combining PDTO drug sensitivity screening and clinicopathological patient characteristics, which improved the prognostic value even further [78]. Other scoring systems have been implemented to translate the results of PDTO in vivo treatment responses to the clinical implementation, like an organoid score based on treatment responses to multiple anticancer treatments, which was significantly correlated to clinical outcomes [79]. Another study showed a 75% PPV and NPV in predicting clinical responses based on a chemotherapy assay on PDTOs in eight different patients with CRC [80]. In another small cohort, a significant correlation between the PDTO response and clinical response to first-line chemotherapies FOLFOX and FOLFIRI was demonstrated, with PDTO outcomes even matching patient prognosis [52]. Additionally, a combined observational study reported a PDTO drug screen accuracy of 83% and 90% for standard-of-care palliative treatment and subsequent-line CRC therapy, respectively [81]. 

The predictive value appears to be treatment-dependent. Retrospective trials demonstrated a significant correlation between the PDTO response and clinical response with oxaliplatin-based treatment [82]. With irinotecan-based therapy, a similar correlation was observed with success rates of predicting tumor responses as high as 80% reported [68,83]. Additionally, a drug screening assay on a cohort of PDTOs of metastasized CRC demonstrated a high predictive value of the Cetuximab response [84], as was described before specifically in rectal cancer [85,86]. All the PDTOs from patients who had a clinical partial response to Cetuximab also had PDTO responding to Cetuximab in the drug sensitivity assay. These results were confirmed in a smaller prospective cohort, where the subsequent-line treatment decision was partly based on the drug screening assay and PDTO genotyping, demonstrating the efficacy of the selected treatment in all cases [84].

However, not all specific chemotherapies yield the same successes. For instance, the correlation between PDTO-predicted responses and clinical efficacy was weaker for 5-FU-based therapy compared to treatments like cisplatin, oxaliplatin, and irinotecan [83,87,88], although one large retrospective study was able to demonstrate a concurrence between PDTO and the clinical response to 5-FU (or its pro-drug capecitabine) and oxaliplatin [89]. 

Focusing on rectal cancer, the neoadjuvant treatment response to irinotecan-based chemotherapy and chemoradiotherapy was predicted with an accuracy of nearly 85% in the latter [90,91,92]. Furthermore, a trial involving PDTOs of locally advanced rectal cancer treated adjuvantly with 5-FU and oxaliplatin therapy both in vitro parallel to in vivo treatment demonstrated a correlation between PDTO drug screens and patient prognosis [93]. Moreover, a correlation between the PDTO and patient tumor response to radiotherapy alone has been repeatedly shown [85,94], demonstrating the usefulness of PDTOs beyond systemic therapy. Recent studies have highlighted the predictive potential of PDTOs for radiotherapy outcomes in CRC. Park et al. conducted a co-clinical trial comparing PDTO responses to irradiation with patients’ clinical radiotherapy outcomes. They found a strong correlation, with a prediction model achieving an accuracy of 81.5% for good responders and an accuracy of 92.1% for poor responders [95]. Yao et al. had a PPV to chemoradiation with 78% sensitivity and 92% specificity [92]. Additionally, Lv et al. reported that patients with radio-sensitive organoids had significantly better metastasis-free and progression-free survival rates after chemoradiotherapy [91]. These findings collectively underscore the potential of using PDTOs to forecast radiotherapy responses. 

Despite the promising results from retrospective trials, the results of two prospective trials in stage IV CRC yielded conflicting results. The SENSOR trial (NL50400.031.14) was a negative study, while the Tumorspheres Colrec study (NCT03251612) was successful. The SENSOR trial, a single-arm, single-center, prospective intervention trial, evaluated the feasibility of using PDTOs to allocate patients for off-label or investigational agents after standard-of-care treatment. Out of 61 patients, only 6 received therapy based on the PDTO screening, indicating a feasibility issue [96]. The Tumorspheres Colrec study, a clinical interventional prospective study in CRC after progression on standard-of-care regimens, based subsequent-line treatment on PDTO drug screening. This study did show the feasibility of the method, demonstrating a significant improvement in 2-month progression-free survival from 31% to 50%, despite a limited PDTO development success rate of 53% [97]. However, they used a culturing medium notably different to other media (see Appendix A). To date, no interventional studies have been published on the use of PDTOs in the first-line treatment of CRC. The results of all the cohort studies have been summarized in Table 1.

Currently, various prospective studies are ongoing to evaluate the feasibility and predictive value of PDTOs in CRC. The EVIDENT trial (NCT05725200) aims to provide individualized systemic therapies to patients with metastasized CRC based on the PDTO drug screening profile, focusing on the feasibility of this method. Another prospective trial is currently including patients with advanced, recurrent, or metastatic CRC to perform drug screening on PDTOs, comparing these results to the clinical outcome (NCT05304741). 

In summary, the predictive utility of PDTOs for precision medicine in CRC has yet to be conclusively proven. While case studies and retrospective studies suggest a high correlation between the PDTO response and clinical responses, prospective studies yield conflicting results and face feasibility challenges. Further rigorous evaluation of PDTOs in clinical settings is urgently needed, with multiple ongoing studies poised to provide deeper insights into their potential benefits for precision medicine in CRC. Moving forward, several issues must be addressed to enhance the clinical utility of PDTOs. Firstly, achieving greater uniformity in PDTO development and analysis is crucial. Additionally, optimizing the media used for cultivating PDTOs, with a focus on developing the optimal medium, is essential for improving their predictive accuracy. Incorporating co-cultures, including CAFs and tumor-associated macrophages, and other components of the TME is important for achieving more accurate responses, even if it may reduce throughput. Furthermore, employing live-cell imaging techniques, as opposed to traditional assays like Cell-TiterGlo^®^, could provide more dynamic and real-time insights into PDTO responses and better capture the complexities of tumor biology. 

### 4.2. Current Limitations and Challenges for Clinical Use of PDTOs 

Whilst data on treatment responses in PDTOs show promise, it is essential to acknowledge that there are various limitations. An important limitation is that these drug screening results stem from isolated tumor tissue. Resistance to CRC treatment can be influenced by the TME, suggesting that more complex PDTO models, such as co-cultures with various cells, could enhance clinical relevance (see Section 2.3). Additionally, the emergence of patient-derived tumor fragments (PDTFs) is a game changer to evaluate the effect of immunotherapy on the TME and tumor cells [98]; however, this goes beyond the scope of this review. 

Cost is another significant hurdle, as obtaining and culturing PDTOs, along with conducting treatment assays, incur substantial expenses [73]. Of course, the potential cost savings from avoiding futile treatments must also be taken into account here. Consequently, the cost-effectiveness of this approach for routine screening may be limited, possibly restricting its use to specific cases, such as therapy-resistant tumors or patients in poor clinical condition. Moreover, the time required for PDTO growth and drug exposure is considerable. To be clinically practical, this timeframe needs to be minimized to ensure the prompt initiation of tailored treatment. Tan et al. demonstrated the feasibility of reporting PDTO results to clinicians within a seven-week timeframe, primarily through assay miniaturization [81]. 

Additionally, the success rate of establishing PDTOs varies widely, ranging from 50% to 90% (see Appendix A) [25,77,85,92,93,97,99,100,101]. More efforts need to be made in optimizing organoid development strategies and interpretation to maximize the utility of PDTOs [81,82]. Moreover, the absence of adequate standardization (e.g., medium) limits comparability. Furthermore, the use of rudimentary screening assays, such as CellTiter-Glo^®^, provides suboptimal readouts. Employing image-based methodologies could offer more comprehensive insights into the nature of the response, distinguishing between cytostatic and cytotoxic effects, while also facilitating the evaluation of intrapatient heterogeneity [102]. A final major drawback is its inability to predict potential toxicities associated with treatments, which could restrict their use. To address this concern, assessing the potential toxicity of therapies using PDTOs on an individual basis has been proposed [103]. However, this approach faces challenges, as it would necessitate biopsies from multiple organs where these toxicities may manifest.

To summarize, relying solely on PDTO-derived treatment decisions is not advisable at present [87]. Nonetheless, PDTO-derived information can complement existing genetic and immunohistochemical analyses, aiding in the customization of each patient’s treatment plan

### 4.3. Are CRC PDTOs Ready for Use at the Forefront?

The current clinical use of PDTOs (and co-cultures) is limited but holds future promise for refining treatment strategies in CRC. By assessing the efficacy of chemotherapy, immunotherapy, targeted therapies, and radiation ex vivo, PDTOs offer a possible means to expedite treatment evaluation and shield patients from the potential adverse effects of ineffective therapies [25,87]. However, several challenges, including success rates, costs, and the need for co-cultures, currently limit the widespread application of PDTOs in clinical practice. Further research is necessary to address these challenges before the broader adoption of PDTOs becomes feasible. In the meantime, a case-based approach complementing the already existing strategies may be utilized to leverage the potential benefits of PDTOs while acknowledging their current limitations.

## 5. Conclusions and Future Directions

We have discussed that, whilst several models of CRC are available for preclinical research, all have significant limitations, like lacking the heterogeneity of tumor tissue and 3D complexity. Therefore, PDTOs are an interesting addition to the current preclinical research modalities, although they come with limitations of their own. Several improvements are at hand to improve the application of PDTOs, like co-cultures and organoids-on-a-chip technology. Moreover, as precision oncology in metastatic CRC is hindered by its focus on single lesion analysis, integrating multiple-lesion PDTOs to capture intertumoral heterogeneity is pivotal. High-throughput drug screening on PDTOs enables effective drug repurposing and synergy screening, advancing CRC treatment significantly. These developments underscore PDTO-based drug discovery as pivotal for accelerating the development of personalized therapies in CRC, leveraging predictive biomarkers like gene mutations and signaling pathway alterations to enhance treatment efficacy.

To date, numerous groups have demonstrated that patient-derived models were able to successfully recapitulate the molecular profiles of CRC and are amenable for functional drug screening and investigations. Clinical data of the correlation between the PDTO response and clinical response are generally positive but are currently mainly based on observational retrospective studies and case reports. So far, few prospective studies have been conducted with varying results. 

All in all, PDTO-based response prediction promises to be complementary to the current clinically used strategies in selecting treatment options. However, the use of PDTOs for assessing CRC treatments faces several limitations for their use in clinical practice. Key challenges include the influence of the TME on drug resistance, high costs, and long culture times. The costs of using PDTOs must be weighed against the savings from possibly avoiding futile treatments. Another limitation is the varying success rate of PDTO establishment, for which the standardization of protocols is pivotal. Additionally, a head-to-head comparison of the various media is urgently needed, not only focusing on generation efficacy but also on histologic, genetic, and transcriptomic similarities with the parental tissue. Moreover, PDTOs cannot predict treatment toxicities, which limits their clinical utility. While considerable work remains, the potential of PDTOs to improve treatment choices and possibly patient prognosis is undeniable, presenting significant promise for future advancements in personalized medicine.

## Figures and Tables

**Figure 1 cancers-16-02671-f001:**
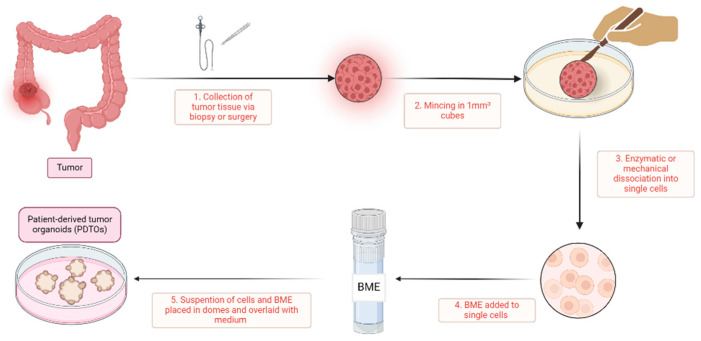
A simplified overview of patient-derived tumor organoid generation. This figure gives an overview of the generation of patient-derived tumor organoids. (1) The process begins with the collection of tumor tissue from patients, obtained via biopsy or surgery; (2) the acquired tissue is then finely minced into small tissue cubes approximately 1 mm³ in size using scalpels; (3) this minced tissue undergoes mechanical or enzymatic digestion, such as with collagenase or dispase, to dissociate it into single cells; (4) these isolated single cells are subsequently mixed with a basal membrane extract (BME), such as Matrigel or Cultrex; and (5) the cell-BME suspension is then plated in domes onto a well-plate or petri dish, which is inverted for half an hour and then overlaid with tumor-specific medium. In these domes, the single cells proliferate and organize into patient-derived tumor organoids. BME, basal membrane extract; PDTOs, patient-derived tumor organoids.

**Table 1 cancers-16-02671-t001:** A summary of the cohort studies studying the correlation between the PDTO-predicted response and clinical response.

Author	Year	Type of Study	Included Patients with Comparison between PDTO and Clinical Outcome	PTDO Establishment Rate *	Disease Stage	Results
Jensen et al. [97]	2023	Prospective	34	53.6%	IV CRC	Two-month PFS improvement of 30% to 50% with treatment based on PDTO DSA resulted
Ooft et al. [96]	2021	Prospective	6	57.4%	IV CRC	Evaluation of treatment based on PDTO DSA discontinued because of drop-out and lack of objective responses
Smabers et al. [82]	2024	Retrospective	23	N/A	IV CRC	PDTO and patient response showed a correlation coefficient of 0.58 for 5-FU and 0.61 for irinotecan- and 0.60 for oxaliplatin-based chemotherapy
Tang et al. [89]	2023	Retrospective	113	78.3%	II–IV CRC	Significant correspondence of PDTO response and patient response to 5-FU + oxaliplatin, with identification of a specific cut-off value for sensitivity prediction
Wang et al. [78]	2023	Retrospective	108	79.4%	IV CRC	AUC value of new PDTO-based drug test prediction model of 0.901 (95% CI, 0.844–0.959)
Lv et al. [91]	2023	Retrospective	107	88%	LARC	Robust predictive ability of PDTO for irinotecan in nCRT (CR: AUC = 0.796, 95% CI = 0.5974–0.9952; pCR: AUC = 0.917, 95% CI = 0.7921–1.0000)
Xue et al. [93]	2023	Retrospective	86	62.3%	LARC	PDTO drug test predicts the benefit of postoperative adjuvant chemotherapy in poor responders to neoadjuvant chemoradiotherapy with an accuracy of 84.8%
Tan et al. [81]	2023	Retrospective	86	N/A	IV CRC	PDTO response prediction with 83% accuracy
Yi et al. [83]	2023	Retrospective	10	N/A	I–IV CRC	Clinical outcomes consistent with drug responses of PDTO in two patients with recurrent disease
Catry et al. [80]	2023	Retrospective	8	61.5–63%	I–IV CRC	Predictive response with 75% sensitivity and specificity of PDTO-based chemograms (25 chemotherapies and targeted therapies)
Martini et al. [84]	2023	Retrospective/prospective	2	55.6–83.9%	IV CRC	Significant PDTO-derived data on drug sensitivity
Geevimaan et al. [68]	2022	Retrospective	42	76–93%	I–IV CRC	70.6% accuracy of oxaliplatin sensitivity test in PDTO
Cho et al. [79]	2022	Retrospective	40	75%	I–IV CRC	Development of “organoid score” based on treatment responses, which correlated with clinical outcomes
Mo et al. [52]	2022	Retrospective	23	80.6%	IV CRC	Significant correlation between PDTO and patient treatment response with AUC of 0.850 for FOLFOX and 0.920 for FOLFIRI
Hsu et al. [94]	2022	Retrospective	16	N/A	I–IV CRC	Distinguishment between poor and good responders of nC(R)T with 100% specificity and 87.5% sensitivity
Wang et al. [77]	2021	Retrospective	45	69.8–80.2%	IV CRC	PDTO prediction of clinical response to chemotherapy with sensitivity of 63%, specificity of 94%, and accuracy of 80%
Park et al. [95]	2021	Retrospective	33	70%	III–IV RC	Positive correlation between radiation response and PDTO responses with AUC of 0.918 and accuracy of 81.5% in good responders and AUC of 0.971 and accuracy of 92.1% in poor responders
Yao et al. [92]	2020	Retrospective	18	85.7%	LARC	84.43% accuracy, 78.01% sensitivity, and 91.97% specificity of concordance between PDTO chemoradiation response and clinical response
Narashiman et al. [88]	2020	Retrospective	2	68%	IV CRC	No correlation between clinical response and PDTO response to FOLFOX
Ooft et al. [87]	2019	Retrospective	29	63.5%	IV CRC	Response prediction of PDTO test in >80% with irinotecan-based therapies; no response prediction with 5-FU + oxaliplatin
Ganesh et al. [85]	2019	Retrospective	19	77%	Stage I–IV RC	PDTO response to radiotherapy corresponded to clinical radiotherapy responses
Vlachogiannis et al. [25]	2018	Retrospective	21	70%	IV (not limited to CRC)	100% sensitivity, 93% specificity, 88% positive predictive value, and 100% negative predictive value of PDTO

5-FU, 5-fluoro-uracil; AUC, area under the curve; CI, confidence interval; CRC, colorectal carcinoma; DSA, drug screening assay; FOLFOX, combination chemotherapy with 5-fluoro-uracil (5-FU) + oxaliplatin; (LA)RC, (locally advanced) rectal cancer; N/A, not applicable; PDTO, patient-derived tumor organoids; PFS, progression free survival; CR, complete respons; pCR, pathologic complete response. * not applicable when an existing biobank was used.

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
