# Peer review of "Advancements in Research and Treatment Applications of Patient-Derived Tumor Organoids in Colorectal Cancer"

_cancers, 2024, doi:10.3390/cancers16152671_

Round 1

Reviewer 1 Report

Comments and Suggestions for Authors

Herein, van der Graaff et al. present an elegant description of an overview of patient-derived organoids for potential treatment development of colorectal cancer. The review is interesting, well written, extensive and timely and will be a great source of information towards the aforementioned elements.

Some specific comments to improve the manuscript:

1.     I suggest the authors shorten some parts, e.g. description of2D and 3D models in the introduction. This is well known and no need to repeat the others.

2.     The authors mention correctly the present of other cell types in organoids – this part could be extended

3.     Chapter 2.2. could profit from a comparative table between the know protocols to depict the most favourable one.

4.     The paragraph with the use of radiotherapy-based treatment optimization based on organoid models could be extended.

5.     Clinical trials are discussed – this is the strongest point of this review. An authors based on this part suggest what is missing or should be developed to make PDOs ever better model for clinical translation?

6.     Please correct multiple typos and empty spaces in the text.

Author Response

  1. I suggest the authors shorten some parts, e.g. description of2D and 3D models in the introduction. This is well known and no need to repeat the others.

We thank the reviewer for his/her comment and agree with the comment. We therefore have shortened this paragraph significantly. We reduced the words on this paragraph from 425 words to 247 words (lines 81-100).

  1. The authors mention correctly the present of other cell types in organoids – this part could be extended.

We agree with the reviewer's suggestion and have expanded this section to include additional details on the presence of another possible cell type in co-cultures with organoids, namely the, tumor-associated macrophages. We have included a study that examines co-cultures with CRC-PDTOs and macrophages and CAFs to highlight their roles and interactions within the organoid model (lines 188-193).

  1. Chapter 2.2. could profit from a comparative table between the know protocols to depict the most favourable one.

We appreciate the reviewer's suggestion regarding the inclusion of a comparative table of known protocols. We already have a supplementary table from our original version to address this topic partially. However, determining the optimal medium extends beyond the scope of our review due to several reasons. Firstly, this subject warrants an independent and comprehensive analysis, as there is currently a lack of sufficient head-to-head comparisons of different media. Secondly, evaluating generation efficiency alone is inadequate for identifying the most favorable medium. A thorough assessment should include immunohistochemical and transcriptomic analyses to ensure that the medium accurately represents the parental tissue and that the inclusion or exclusion of stemness factors does not inadvertently regulate critical pathways. Additionally, it is crucial to perform comparative studies to determine which medium best reflects clinical drug responses. Therefore, we believe that we currently lack the necessary information to definitively identify the most favorable medium. Nonetheless, we will mention this limitation and the need for further research in our 'Further Research' section (lines 597-600).

  1. The paragraph with the use of radiotherapy-based treatment optimization based on organoid models could be extended.

We thank the reviewer for the valuable feedback and agree that the paragraph on radiotherapy-based treatment optimization using organoid models was rather brief. To address this, we have expanded the discussion by incorporating additional information on specific studies related to radiotherapy (lines 469-477).

  1. Clinical trials are discussed – this is the strongest point of this review. An authors based on this part suggest what is missing or should be developed to make PDOs ever better model for clinical translation?

We appreciate the reviewer's positive feedback on the discussion of clinical trials, which we agree is a strong aspect of our review. In response to the suggestion, we have added our insights at the end of this section. Key areas for development to enhance PDTO for clinical translation include standardization of protocols, optimization of culturing media, incorporation of co-cultures, and the adoption of advanced imaging techniques such as live-cell imaging, rather than relying solely on assays like CellTiterGlo® (lines 515-524).

  1. Please correct multiple typos and empty spaces in the text.

We thank the reviewer for pointing this out. We thoroughly assessed the manuscript and corrected the typos and empty spaces.

Reviewer 2 Report

Comments and Suggestions for Authors

       In this manuscript entitled “Advancements in Research and Treatment Applications of Patient-Derived Tumor Organoids in Colorectal Cancer”, the author reviewed the application of patient-derived tumor organoids (PDTOs), three-dimensional, self-organizing models derived from patient tumor samples, as screening tools for drug resistance in CRC. PDTOs offer unique advantages over traditional models by recapitulating the tumor architecture, cellular heterogeneity, genomic landscape and are a valuable ex vivo predictive drug screening tool. This review provides an overview of the current literature surrounding the use of PDTOs as instrument for predicting therapy responses in CRC, hence offering a new approach to identify existing drugs effective against drug-resistant CRC. Overall, this review is well written. However, the following issues should be addressed before the acceptance of this paper.

1)     The advantages of PDTOs should be better illustrated to emphasize its advantages.

2)     How should the researchers utilized novel techniques such as RNA-sequencing and single cell sequencing to better identify and categorize the type of the obtained tumor tissues?

3)     More references related to the application of patient derived tumor models should be added in this review. (Advanced Functional Materials 2023, 33 (49), 2307013). 

Author Response

  • The advantages of PDTOs should be better illustrated to emphasize its advantages.

We thank the reviewer for the feedback. While we believe that the advantages of patient-derived tumor PDTOs are well-established, we have added a sentence to further emphasize their benefits and enhance the focus on their advantages (lines 61-63).

  • How should the researchers utilized novel techniques such as RNA-sequencing and single cell sequencing to better identify and categorize the type of the obtained tumor tissues?

We thank the reviewer for highlighting this point. To address the clarity of how novel techniques like RNA-sequencing and single-cell sequencing can be utilized, we have added a paragraph discussing these methods in detail. Additionally, we have included a study on the characterization of CRC PDTOs using scRNAseq to provide further context and examples (lines 239-251).

  • More references related to the application of patient derived tumor models should be added in this review. (Advanced Functional Materials 2023, 33 (49), 2307013

We respectfully disagree with the suggestion to include additional references beyond those already cited. We believe that the references included in our review are comprehensive and adequately cover the key aspects of patient-derived tumor models relevant to our discussion. Adding more references without specific relevance may not enhance the review's focus or clarity. Our aim has been to provide a balanced and targeted overview, and we feel that the current references appropriately support the content of our review. Moreover, the suggested reference focuses on a specific approach to enhance tumor penetration and STING activation in osteosarcoma, which does not directly pertain to the organoid models discussed in our review
